# A New Theory about Interfacial Proton Diffusion Revisited: The Commonly Accepted Laws of Electrostatics and Diffusion Prevail

**DOI:** 10.3390/biom13111641

**Published:** 2023-11-12

**Authors:** Denis G. Knyazev, Todd P. Silverstein, Stefania Brescia, Anna Maznichenko, Peter Pohl

**Affiliations:** 1Institute of Biophysics, Johannes Kepler University, 4020 Linz, Austria; denis.knyazev@jku.at (D.G.K.); stefania.brescia@jku.at (S.B.); anna.maznichenko@jku.at (A.M.); 2Chemistry Department, Willamette University, Salem, OR 97301, USA; tsilvers@willamette.edu

**Keywords:** PMF, surface proton, TELP, interfacial water, proton diffusion

## Abstract

The high propensity of protons to stay at interfaces has attracted much attention over the decades. It enables long-range interfacial proton diffusion without relying on titratable residues or electrostatic attraction. As a result, various phenomena manifest themselves, ranging from spillover in material sciences to local proton circuits between proton pumps and ATP synthases in bioenergetics. In an attempt to replace all existing theoretical and experimental insight into the origin of protons’ preference for interfaces, TELP, the “Transmembrane Electrostatically-Localized Protons” hypothesis, has been proposed. The TELP hypothesis envisions static H^+^ and OH^−^ layers on opposite sides of interfaces that are up to 75 µm thick. Yet, the separation at which the electrostatic interaction between two elementary charges is comparable in magnitude to the thermal energy is more than two orders of magnitude smaller and, as a result, the H^+^ and OH^−^ layers cannot mutually stabilize each other, rendering proton accumulation at the interface energetically unfavorable. We show that (i) the law of electroneutrality, (ii) Fick’s law of diffusion, and (iii) Coulomb’s law prevail. Using them does not hinder but helps to interpret previously published experimental results, and also helps us understand the high entropy release barrier enabling long-range proton diffusion along the membrane surface.

## 1. The Surface Proton Concept in Bioenergetics

In bioenergetic membranes, proton pumps energize the inner membrane by actively translocating protonic charges from the matrix to the intermembrane space [1,2]. The F_1_F_0_ ATP synthase is then driven by the transmembrane difference in the electrochemical potential of protons, ∆*µ*_H+_, between the aqueous phase on the inner (N) side of the bioenergetic membrane (the matrix in mitochondria) and the aqueous phase on the outer (P) side (usually the cytoplasm) [3]. The transmembrane electrical potential Δψ at the inner mitochondrial membrane and the difference in pH values, ∆pH, between the aqueous phases on the N and P sides give rise to ∆*µ*_H+_:(1)$$\Delta \mu_{H^{+}}=zF\Delta \psi +RTln\frac{{\left[H\right]}_{N}}{{\left[H\right]}_{P}}\approx zF\Delta \psi \;–2.3RT\Delta {pH}_{b}$$

$\Delta \mu_{H^{+}}$ is in units of kJ/mol (or kcal/mol); *z* = +1, the charge of the proton; *F* = 96.4853 kJ/mol/V, Faraday’s constant; *R* = 0.008314 kJ/mol/K, the universal gas law constant; *T* is the temperature in K; and ∆ is defined for import, from the outer (P) side to the inner (N) side; ∆pH_b_ = bulk phase pH(N) − pH(P). In almost all bioenergetic membrane systems, the electrical potential, ψ, inside the cell (or mitochondrion) is negative relative to the outside, and the pH inside the cell (or mitochondrion) is higher relative to the outside (i.e., more alkaline); alkaliphilic bacteria are a notable exception to this generalization. Some authors prefer to express ∆*µ*_H+_ in units of volts, as the protonmotive force, pmf, where [4]:(2)$$pmf\equiv -\frac{\Delta \mu_{H^{+}}}{zF}$$

Δψ and ∆pH are distinct contributors to ∆*µ*_H+_. Initially, ∆*µ*_H+_ is established by proton pumping through electron-transfer-chain coupling sites; this initial uncompensated export of positive charge establishes both the inside negative Δψ and the positive ∆pH (alkaline inside). However, the existence of other ion transporters in the membrane can serve to decouple Δψ from ∆pH. For example, an uncompensated anion exporter or cation importer would collapse Δψ while leaving ∆pH intact. On the contrary, an electroneutral proton/anion symporter or proton cation antiporter in the membrane would collapse ∆pH while maintaining Δψ. This explains why, in mitochondria, the main component of ∆*µ*_H+_ can be Δψ, whereas, in chloroplasts, it is always ∆pH [5].

Both Δψ and ∆pH may spread along the membrane [6]. The spreading of Δψ is due to the high electric conductance of the media on both sides of the membrane and the low conductance of the membrane. In eukaryotes it has been observed along mitochondrial filaments [7], and in prokaryotes along trichomes of filamentous cyanobacteria [8]. ∆pH spreading is aided by the high rate of H^+^ diffusion. Thus, a ∆*µ*_H+_ generated in a membrane patch can, in principle, be transduced into work when used in a distant patch. These considerations do not preclude inhomogeneities of both Δψ and ∆pH at different points within mitochondria [9,10] or even planar lipid bilayers [11].

Protons attain the highest rate of diffusion when not bound to buffer molecules. Once bound, their diffusion rate decreases—often by an order of magnitude or more [12]. Moreover, protons carried by buffer molecules are not accounted for by Equation (1). These considerations led to the question of whether there is an H^+^ pathway along the membrane surface that does not involve buffer molecules. The answer is of particular interest for bioenergetics as it clarifies how the protons reach the membrane site where they are consumed for ATP production (proton sink) from the site where they are released (proton source) [13] or how mild mitochondrial uncoupling may work [14,15].

The proton pathway has been widely debated for decades [16,17,18]. The creator of the chemiosmotic theory, Nobel Prize winner Mitchell, assumed rapid proton equilibration throughout the mitochondrial matrix [19,20,21]. This view was generally prioritized before the mid-1990s. The first evidence obtained to the contrary [22,23,24] was largely dismissed. One of the reasons was that the experiments [23,24] suffered from having the proton source in the bulk instead of at the interface [25]. Yet, experimental evidence accumulated in the 1980s and 1990s indicated a retarded release of the proton from the membrane interface to the bulk due to the presence of a significant energetic barrier [26,27,28,29,30,31,32]. More recent demonstrations involved photo acids [33] or a system containing reconstituted cytochrome bo_3_ (the source) and F_1_F_0_ ATP synthase (the sink). ATP synthase efficiency was maximized when the source and the sink were within the range of 2D proton diffusion [34].

Electrostatic attraction of the positively charged protons to negatively charged membranes appealed as a possible mechanism for proton attraction. The electrostatic Gouy–Chapman model of the diffuse double layer [35,36,37] provides the theoretical framework predicting higher proton concentration adjacent to a negatively charged membrane compared to the bulk phase. Experimentally assessing the equilibrium difference in free energies, ΔG, for the proton populations at the membrane interface and in the bulk phase, requires direct measurements of the proton concentration distribution in the first few nanometers adjacent to a membrane. An elegant study positioned pH sensors at different distances from the membrane and confirmed that proton distribution can indeed be described with the Gouy–Chapman model [38] (Figure 1B). The characteristic distance, λ_D_, corresponded to the Debye length of the diffuse double layer. Other reports of equilibrium proton concentrations at the surface of the mitochondrial inner membrane revealed equally slight pH differences [39,40]. In yet another approach, the interfacial pH was estimated from the protonation kinetics of a fluorescent pH sensor in lipid vesicle membranes. It also corroborated the conclusion that negative charges increase the interfacial proton concentration [41].

As an immediate consequence of the increased proton surface concentration, the apparent pK_a_ value of interfacial residues differs from the respective bulk-phase values. For example, the pK_a_ of oleic acid increases by several pH units at the lipid–water interface as compared with the pK_a_ of the carboxyl group in bulk [42]. The pKa shift depends on the ionic strength of the solution, as has been shown with phosphatidylserine [43]. The effect is due to alterations in the electrostatic screening of the surface charges. And yet, it is difficult to predict the pK_a_ shift from the resulting membrane surface potential, because the pK_a_ of long-chain fatty acids is also determined by the difference in the energetic cost of burying the deprotonated anionic carboxylate vs. the protonated neutral COOH group in the hydrophobic interior of the lipid bilayer [44].

Strikingly, proton migration along the lipid bilayer surface does not require negatively charged bilayers. Protons are perfectly capable of diffusing tens of micrometers on top of uncharged membranes [45]. They move from a membrane patch where the photoconversion of a membrane-bound caged compound releases them to a distant measurement spot at a rate that exceeds by several-fold the rate of mobile buffer movement. The protons at the front may thus appear electrostatically unbalanced, yet their number is negligibly small compared to the total number of ions already at the interface. Consequently, the forces these excess protons may create across the membrane are negligible. MD simulations of excess protons initially did not yield an extended range of diffusion [46]. Using a method to include the Grotthuss proton-shuttling mechanism in classical molecular dynamics simulations, or the third generation of the multistate empirical valence bond model, showed more extended diffusion spans and interactions with the polar carbonyl oxygens [47] and phosphate moieties [48]. Yet µm long diffusion spans have not been observed in silico.

Long-range interfacial proton diffusion would be impossible without a significant barrier opposing proton surface-to-bulk release. This kinetic barrier must be distinct from the ΔG value derived from equilibrium measurements or predicted by the Gouy–Chapman theory (Figure 1B). Importantly, titratable residues are also not required to create the proton release barrier. The long-range interfacial proton migration persisted on lipid bilayers lacking titratable residues [49]. Simple calculations also showed that the measured diffusion rates would not be attainable if titratable residues were involved [49] because the rate of proton release from titratable groups is too slow for moieties with a basic pKa [50]. This is precisely the reason why diffusion constants predicted from protonation rates [51] are orders of magnitude too slow [49].

Experiments on the decane–water interface confirmed that titratable residues are not required for interfacial proton diffusion [52]. Proton movement could be adequately described with a nonequilibrium model of 2D diffusion, assuming an irreversible proton release reaction from the surface. The model had first been introduced by proposing that the traveling wave of excess protons might saturate the titratable residues at the interface, thereby preventing these residues from inhibiting proton migration [53]. The out-of-equilibrium description of interfacial proton diffusion [53,54] allowed determining the Gibbs activation energy barrier $\Delta G^{‡}$ (Figure 1A) that opposes proton release from the surface to be roughly equal to 30 *kT*. In contrast to previously introduced equilibrium models [49,55], this nonequilibrium model appeared to describe the experimental results much more accurately.

Significantly, a clear understanding of the origin of $\Delta G^{‡}$ has not yet evolved. What is clear is that the barrier is mainly entropic in origin [54,56], i.e., somehow associated with the ordering of water at interfaces. Conceivably, the extended network of hydrogen bonds at the membrane–water interface [57,58] plays a role. Ab initio simulations aimed at a deeper understanding of the above-cited measurements of proton transfer along the decane–water interface confirmed that water orientation at the interface contributes to the proton release barrier [52]. In itself, the role of ordered water is not surprising as even proton transfer in bulk water is a complex phenomenon involving higher levels of transient water–proton structures [59,60,61]. Multistate empirical valence bond model simulations also show a surface preference for the excess proton at water–hydrophobic interfaces. Yet, the hydrated proton structures were stabilized due to the “amphiphilic” nature of the hydrated excess proton [62]. Whereas there is little doubt that the amphiphilic nature contributes to the overall barrier, the sheer size of $\Delta G^{‡}$ implies that additional contributors are at work on the surface of lipid membranes.

The most likely reason for the discrepancy between the experiments and simulations is the differences in the physicochemical properties between interfacial water and bulk water [63,64,65]. Nowadays, ab initio simulations could provide a solution, yet the computational effort is enormous. The only respective ab initio simulation known to date considered a lone excess proton dissolved in water on top of a bilayer of 16 lipids [66].

The lack of a molecular understanding of $\Delta G^{‡}$’s origin provoked alternative approaches to explain the underlying experiments. The first of these attempts led to a quasi-equilibrium model [67]. It introduces a transition layer between the interface and the bulk phase, from which the released proton could rebind to the interface. The model successfully explains some aspects of experiments with photoacids [68], yet it fails to adequately reflect experiments with long-range proton diffusion at different temperatures [54].

A second attempt has been proposed by James W. Lee. He recently published a series of papers presenting his “transmembrane electrostatically localized protons” (TELP) hypothesis [69,70,71,72,73,74,75]. Here, the bilayer membrane is modeled as a simple capacitor. The head group–fatty acid interfaces at either side of the bilayer membrane represent the bottoms of two capacitor plates; a layer of thickness, *l*, extending from this interface into the aqueous phase forms the rest of the plate. The bilayer interior serves as the space between the two plates. Applying the equation of a capacitator, Lee predicted that the number of protons, *n*_H+_, that accumulate in an area, *A*, adjacent to the P surface of the inner mitochondrial membrane is given by Equation (3):(3)$$\frac{Q}{A}=\frac{{e\;n}_{H^{+}}\;\;}{A}=-\frac{C}{A}\Delta \psi \;;$$
where *e* is the elementary charge. Equation (3) ignores all ions except protons. To stabilize *n*_H+_ protons in the surface layer, Lee proposed that *n*_OH−_ hydroxide anions accumulate at the matrix (N) side of the membrane. Substituting $C=\varepsilon_{0}\varepsilon A/d$, where *d* is the distance between the two “capacitor” plates, and accounting for all protons in a layer of thickness l, Equation (3) transforms into:(4)$${\left[H^{+}\right]}_{0}=-\frac{\varepsilon_{0}\varepsilon }{dl}\frac{\Delta \psi }{F}\;;$$

Note that the bulk aqueous phase begins abruptly at any distance > *l.* Also, the TELP hypothesis provides a static picture. ${\left[H^{+}\right]}_{0}$ depends only on the specific membrane capacitance and Δψ (Figure 1C). Since time does not enter Equations (3) and (4), the TELP hypothesis is unable to describe processes where the concentration of protons changes with time. Even so, Lee attempted to employ the TELP hypothesis [74] to interpret the results of our proton diffusion experiments [52], where the proton moves along the decane–water interface between a release spot and an observation spot. Being unable to provide a fit of Equation (4) to our data, i.e., to a ${\left[H^{+}\right]}_{0}$ that changes with time, he simply replotted our figures while claiming to have modified them [74]. In his paper, Lee also opined that electroneutrality does not hold due to the difference in diffusion coefficients of proton and chloride ions, so the moving front of chloride ions lags behind that of protons. Sadly, the paper did not advance our knowledge concerning the origin of protons’ affinity to interfaces.

**Figure 1 biomolecules-13-01641-f001:**
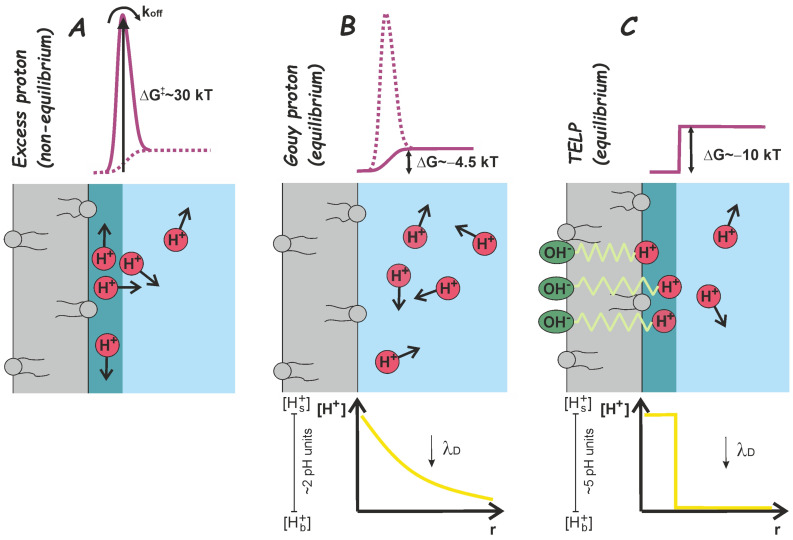
The surface and bulk proton concentrations differ: (**A**) $\Delta G^{‡}$, the activation free energy of proton surface-to-bulk release accounts for the retarded release of excess protons (i.e., protons not accounted for by an equilibrium description). Such barriers are not specific to biological membranes, as long-range diffusion has also been observed adjacent to inorganic substrates [76] and at the water–nonpolar liquid interface. $\Delta G^{‡}$ cannot be used to predict the difference between interfacial and bulk pH values since it characterizes an out-of-equilibrium state (in contrast to B). Importantly, ΔG^‡^ does not permit estimating the equilibrium proton concentration difference between the membrane surface and the bulk phase. Such assessments require knowledge of ΔG. (**B**) At equilibrium, the higher proton concentration at the lipid membrane–water interface may be due to the negative surface potential of natural lipid membranes, as described by the Gouy–Chapman theory. The H^+^ concentration decreases exponentially with increasing distance from the surface with the Debye length, $\lambda_{D}$ [38]. Titratable residues at the interface, such as ethanolamine lipid headgroups, may provide a surface buffer for protons [77,78]. However, proton release kinetics have only a tiny effect on the migration kinetics between the proton source and sink. The purple line sketches the free energy profile of protons normal to the interface. Solid lines indicate which part of the profile was measured experimentally. The yellow line shows the corresponding equilibrium proton concentration profile. (**C**) Lee’s TELP hypothesis postulates that the proton concentration adjacent to the outer surface of the inner mitochondrial membrane is governed by the transmembrane potential, Δψ. The TELP hypothesis assumes that hydroxide anions adsorb on the matrix side of the membrane. The protons in the surface layer of thickness *l* have no counterions and do not form a double layer—in contrast to the Gouy–Chapman theory. The TELP hypothesis does not explain why protons do not interact with hydroxides from the same side or why Δψ does not alter the concentration of other ions in the immediate membrane vicinity. We plot the equilibrium concentration and energy profiles for H^+^ using Lee’s estimations of interfacial pH drop [70].

As we show below, Lee’s TELP (transmembrane electrostatically localized proton) hypothesis does not aid in understanding the molecular mechanisms behind the proton surface-to-bulk release barrier. In contrast to Lee’s most recent assertions [74,75], both published in *Biophysical Chemistry*, his static capacitor-based TELP hypothesis is not designed to describe long-range interfacial proton migration [27,52,79]. In fact, the TELP hypothesis adds only confusion to a complicated puzzle that has already occupied scientists for decades and thus has already provoked criticism [80]. The field requires non-biased (ab initio) simulations of protons in interfacial water on top of lipid bilayers. We expect such focused theoretical input will unravel the contributions to the immense entropic barrier to proton release.

## 2. Results and Discussion

### 2.1. Postulates of the TELP Hypothesis

The postulates of the TELP hypothesis are reproduced below in *italic type* followed by our critique in normal type.

**Postulate** **1:***The* Δψ*-creating protons in the localized interfacial layer do not have counter ions in the same compartment [81].*

Since the TELP hypothesis assumes equilibrium, this postulate violates the principle of electroneutrality. Lee’s assertion that interactions with counterions (OH^−^) at one side of the membrane would electrostatically stabilize H^+^ at the other side is wrong. This conclusion is the immediate consequence of the experimental conditions Lee used in his “biomimetic” electrolysis experiments [81]. There, 75 µm thick Teflon disks served as membrane “biomimetics”. Teflon has a permittivity of ε = 2, resulting in a Coulomb interaction energy between two elementary charges, *e*, being equal to the thermal energy *kT* if the separation between the two charges is *l*_B_ = 280 Å, where *l*_B_ is called the Bjerrum length. The textbook equation defines *l*_B_ as [82]:(5)$$l_{B}=\frac{e^{2}}{kT}\frac{1}{4\pi \varepsilon \varepsilon_{0}}$$

Since *l*_B_ = 28 nm is more than three orders of magnitudes smaller than the ion separation distance of 75 µm in the experiments, the electrostatic attraction between H^+^ and OH^−^ on the two sides of the Teflon disk is negligible. Thus, TELP’s postulated equilibrium accumulation of protons at the Teflon disk surface violates the principle of electroneutrality.

**Postulate** **2:**
*The*

 Δψ

*-creating protons are distinct from the protons and ions in the diffuse double layer, which can be described by the Gouy–Chapman theory because “…this double layer always exists at all times during light and dark conditions even if the proton motive force is zero” [81]. Accordingly, λ_D_ “cannot be used to estimate the thickness of the localized excess proton layer because the equation can be applied only to charge balanced solutions including one-to-one electrolyte solutions such as NaCl”. [81].*


The distinction between Δψ-creating protons vs. the protons in the diffuse double layer is unfounded. Excluding part of the protons from the diffusional exchange with adjacent protons violates Fick’s law of diffusion. What is more, the very formation of a localized excess proton layer violates Fick’s law of diffusion. Since interfacial protons cannot interact with counterions on the opposite side of the membrane (see above), they must distribute evenly throughout the compartment.

Commonly, proton pumping is thought to generate the potential at the inner mitochondrial membrane [83], i.e., the selective transfer of positive charges creates Δψ (Figure 2A). In contrast, Lee claims: 

**Postulate** **3:**
*Electrostatic forces preferentially drive protons to the localized layer, where their concentration is much higher, i.e., their mutual repulsion in the bulk solution commits them to the membrane surface (Figure 2). A similar repulsion of hydroxide ions in the mitochondrial matrix leads to OH^−^ accumulation at the other side of the membrane. Within these layers, Lee assumed that H^+^ (or OH^−^) exceeds Na^+^ and K^+^ (or other anions) by a factor of 10^8^ [81].*


The origin of such repulsion specificity remains enigmatic. Because sizes of the hydrated H^+^, Na^+^, and K^+^ cations are of the same order of magnitude (5–10 Å), the idea that the positive charge of K^+^ or Na^+^ is many orders of magnitude less susceptible to electrostatic forces violates Coulomb’s law.

### 2.2. Lee Supports These Postulates with Results from His Electrolysis Experiments

In his “key” experiment, Lee used 200 V to produce H^+^ by electrolysis [81]. Two 75 µm thick Teflon septa covered by 25 µm thick aluminum films on both sides divided the chamber into three compartments and separated the anode and cathode. An additional aluminum film was placed in the central chamber. Lee observed the corrosion of some of the aluminum films and attributed this to H^+^ accumulation on their surfaces. He imagined the raised proton concentration at these surfaces to be driven by charge repulsion in the bulk solutions. As discussed above, this model violates Coulomb’s law and Fick’s law. Instead, the high electrolytic voltage (=200 V) charges the metallic surfaces between the anode and cathode. Because protons represent the major charge carriers in these experiments, they are bound to accumulate at these surfaces, thereby corroding the aluminum films they come in contact with.Lee noticed that the aluminum film corrosion depended on the solution’s ionic strength. From the K^+^ or Na^+^ concentration required to reduce the corrosion of the Al foil, he calculated an H^+^/cation exchange factor on the order of 10^8^. According to the TELP hypothesis, this factor signifies the number of cations necessary to exchange for one proton in the surface layer. Yet, the well-known dependence of electrolysis on solution ionic strength would provide a more realistic explanation. It has the great advantage of not assigning specificity to electrostatic interactions beyond that of the sign and size of the charges. Moreover, this more realistic interpretation agrees with the observation that electrolysis can be used to measure the ionic strength of the solution [84]. In addition, we have shown [85] that thermodynamic predictions using the H^+^/cation exchange factor of ≈ 10^8^ are not corroborated by the data presented in the Saeed and Lee paper [81].Lee assumed that aluminum films exposed to a 200 V electric field in an electrolysis chamber are adequate models for studying the interactions of protons with the inner mitochondrial membranes. We disagree as relevant features like hydrophobicity, interfacial water structure, and proton permeability differ between the model and object.

### 2.3. Lee Claims That the TELP Hypothesis Explains Previously Obtained Experimental Data on Proton Interfacial Diffusion

Long-range proton migration between a proton source and a sink can only occur if attractive forces keep the proton at the interface. These attractive forces have long been attributed to titratable residues at the interface [51,86]. Since proton release from such residues is too slow to account for the observed surface diffusion coefficient [45], the titratable residue hypothesis had to be discarded [49]. To demonstrate surface diffusion in the absence of titratable residues, we previously tested a minimalistic model [52] consisting of an interface between an apolar hydrophobic phase (*n*-decane) and an aqueous phase (Figure 3). In our experiments, we injected HCl in one spot at the interface. We observed H^+^ arrival at a distant spot (Figure 3A). We found that the diffusion span, on the order of tens of micrometers, and the surface diffusion coefficient were both similar to the values observed for the lipid–water interface [45,49,56].

Recently, Lee claimed that his TELP model may be used to explain our experimental results (Figure 3) and provide new insight [74]. Yet, the claim is unjustified. The TELP model does not apply to our experimental system for several reasons:The TELP model is static, so it does not account for transient changes in proton concentration at the interface.The TELP hypothesis requires a charge imbalance in two compartments separated by a membrane. Otherwise, there would be no repulsion between the protons (or OH^−^), which, according to Lee, is a prerequisite for proton (or OH^−^) accumulation at the interface. Our experimental settings do not fulfill this requirement (Figure 3). The microinjection of HCl does not cause a charge imbalance, i.e., charge neutrality is always strictly maintained.The TELP model envisions interfacial protons stabilized by a potential between two capacitator plates where the voltage between these plates is proportional to the number of surface protons (Equation (3). Our experimental system (the decane–water monolayer interface) has only a single “plate” [52]. Hence the TELP mechanism of Δψ generation [69] cannot work.

Consequently, our earlier conclusion that strongly associates the origin of $\Delta G^{‡}$ with the peculiar structure of water at interfaces [45,66,87] remains unaltered. The corresponding entropic barrier is far larger than any enthalpy contribution to $\Delta G^{‡}$ (Figure 4) [54,56]. For uncharged membranes, the size of the enthalpic contribution is comparable with the energy of a single hydrogen bond. For charged membranes, the attraction to the negative surface charge may result in a somewhat increased enthalpic constituent to $\Delta G^{‡}$ [56]. In any case, the enthalpic contribution is so small that it does not hamper interfacial proton diffusion. Even for positively charged membranes, the diffusion span may amount to tens of micrometers [56]. Proton binding to titratable interfacial residues does not constitute a mandatory part of the release barrier [49]. As a result, interfacial proton travel is not limited by the proton’s residence time at such residues.

To summarize, we agree with Lee that the apparent proton affinity to the membrane–water, or in general, to the hydrophilic–hydrophobic liquid interface, is not explained. However, disregarding the complexity of the interfacial proton diffusion phenomenon renders the TELP hypothesis unfit for this task. We expect that the growth in calculation power observed in the last decade will soon enable ab initio simulations complex enough to greatly improve our understanding of interfacial proton diffusion.

## Figures and Tables

**Figure 2 biomolecules-13-01641-f002:**
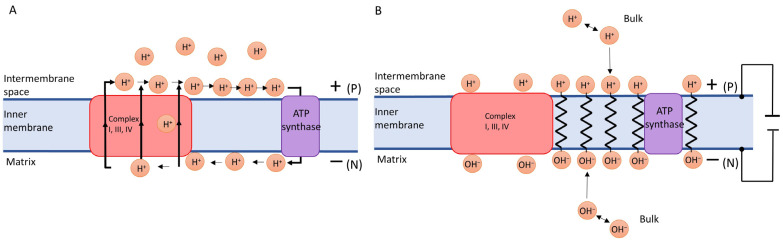
Origin of the potential at the inner mitochondrial membrane. (**A**) Δψ is generated by proton pumps [83]. Complexes I, III, and IV transfer positively charged protons from the matrix to the intermembrane space, thereby causing charge separation. The protons then may travel between these pumps and a proton sink, e.g., uncoupling proteins or the ATP synthase, along the outer membrane surface. After being subsequently released at the N surface, interfacial proton migration may occur again, i.e., the protons may travel in the opposite direction. (**B**) Lee’s TELP hypothesis stipulates that proton pumping into the intermembrane space produces excess protons there and leaves excess hydroxide anions in the matrix. Mutual repulsion of the excess charges in both compartments results in their accumulation at both sides of the inner membrane, where they are supposedly stabilized by mutual attraction across the membrane. The accumulated ions represent the charges on capacitor plates giving rise to Δψ.

**Figure 3 biomolecules-13-01641-f003:**
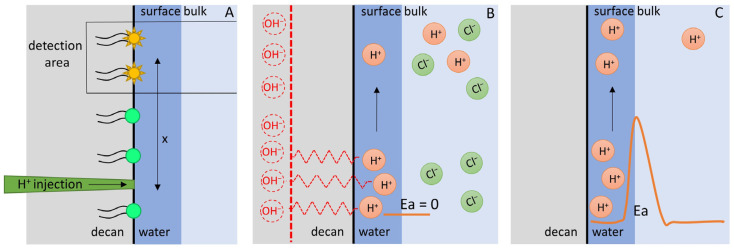
Proton migration along the decane–water interface. (**A**) In an effort to find a minimalistic system that would enable interfacial proton diffusion, we injected protons at a small spot at the decane–water monolayer interface [52]. Observing the proton concentration change at a distant detection site allowed the calculation of the diffusion coefficient and the surface-to-bulk release constant. (**B**) Lee’s TELP hypothesis stipulates that only protons deprived of counter-charges may enter the interfacial layer and that these protons are stabilized by an equal number of OH^−^ in the opposite interfacial layer (dotted lines and letters). Yet, in our experiments, all protons have counterions (Cl^−^), and the stabilizing OH^−^ are missing—as is the second interface. The lack of a second capacitator plate renders the capacitator model inapplicable. As indicated by the arrow, the absence of Δψ allows all protons to leave the membrane in a barrier-free fashion. (**C**) In contrast to the predictions of the TELP hypothesis, we observed long-range interfacial proton migration [52]. Consequently, a significant energy barrier that prevents proton surface-to-bulk release must exist. Later experiments showed that the barrier is mainly entropic in nature [54].

**Figure 4 biomolecules-13-01641-f004:**
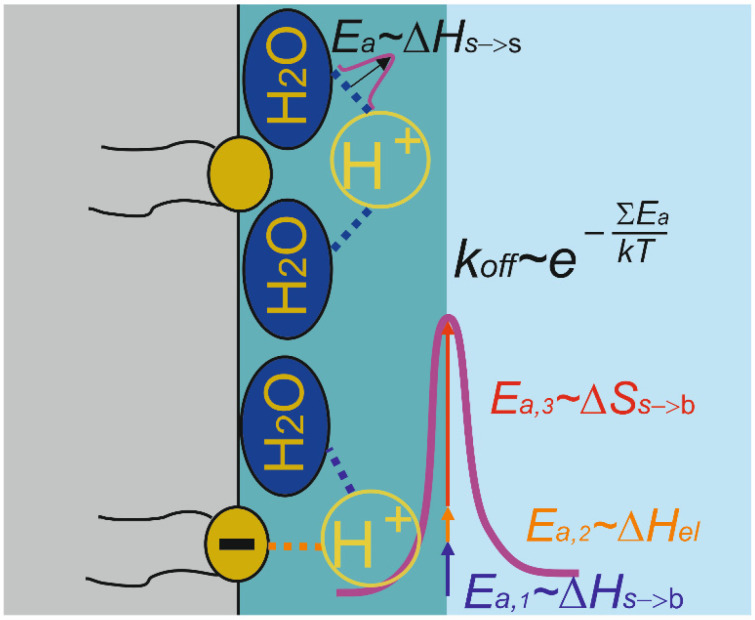
The proton surface-to-bulk release barrier. Mounting evidence indicates that the peculiar structure of water at interfaces generates an entropic barrier (*E*_a,3_) that constitutes the major contribution to $\Delta G^{‡}$ (∑*E*_a_). Minor enthalpic contributions come from hydrogen bonding (*E*_a,1_) and electrostatic interactions with charged interfacial residues (*E*_a,2_). Accordingly, the proton surface-to-bulk release constant *k*_off_ scales with $\Delta G^{‡}$, i.e., *E*_a,1_ + *E*_a,2_ + *E*_a,3_. Proton diffusion along the membrane surface is favored by the fact that the barrier *E*_a_ for moving from one water molecule at the surface to the next is much smaller than that for releasing the surface proton into the bulk phase.

## Data Availability

No new datasets were generated.

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
