# Peer review of "A New Theory about Interfacial Proton Diffusion Revisited: The Commonly Accepted Laws of Electrostatics and Diffusion Prevail"

_biomolecules, 2023, doi:10.3390/biom13111641_

Round 1
Reviewer 1 Report
Comments and Suggestions for Authors
Knyazev et al. reported a new theory about interfacial proton diffusion by comparing it with TELP proposed by Lee. The new theory has been proposed to explain interfacial proton diffusion, emphasizing the relevance of the well-established laws of electrostatics and diffusion. According to this theory, the commonly accepted principles of electrostatic interactions between charged species and the process of diffusion play a significant role in describing the phenomenon.
The theory suggests that the attraction or repulsion between protons and charged interfaces, such as membranes or interfacial residues, influences the mobility of protons at the interface. The electrostatic forces between the charges contribute to the overall energy landscape and affect the diffusion of protons across the interface.
Furthermore, the theory acknowledges the importance of the diffusion process itself. It recognizes that diffusion is a fundamental mechanism for the movement of particles, including protons, and that it is governed by well-established laws. Factors such as concentration gradients, diffusion coefficients, and intermolecular interactions contribute to the overall diffusion behavior of protons at interfaces.
By considering the interplay between electrostatics and diffusion, this new theory provides a framework to understand interfacial proton diffusion without relying on specific hypotheses or assumptions about proton binding or release at interfacial residues.
However, further experimental investigations and theoretical studies are needed to validate and refine this new theory and to fully grasp the intricacies of interfacial proton diffusion.
Hence, I have a neutral opinion on this article.
Reviewer 2 Report
Comments and Suggestions for Authors
I appreciate the manuscript very much as we need critical evaluation of published data. Definitively it should be published.
However, in order to maximize the impact I suggest to change the writing style. It doesn´t help much to emphasize repeatedly that the TELP theory is wrong. For other readers to follow the authors should rather start with a set of general definition of the chemical potential as ions are coupled via the potential. (BTW, if the author use 2.3 they could also simplify the natural constants).
Overall I wish to see a better introduction into the problem (I know the author knows well the problem but the general reader or reviewer is not so quick). Charge (and proton) distribution at the interface following Gouy-Chapman act mainly in the aqueous phase, the ions are mobile and quickly redistribute. Lipids contain often ionizable parts which cause local effects (surface pKa), I m not sure inasmuch macroscopic theories are accurate enough to cover the effect originating from the first layers (below Debye length) and the current experiments likely measure rather equilibrium values. For these ions charge neutrality holds.
i m convinced that a clear statement of the problem will support the community
Excess ions create forces across the membrane and are unbalance but are only a very few compared to those already at the interface.
I suggest the authors to extend their discussion and to distinguish the various limits: excess charges vs. excess, local effects below the Debye length and maybe include some discussions on Molecular Dynamics which might be able to cover such effects.
Round 2
Reviewer 2 Report
Comments and Suggestions for Authors
I m satisfied with the modifications. Ths ms. should be published